# Severe Acute Myocarditis after the Third (Booster) Dose of mRNA COVID-19 Vaccination

**DOI:** 10.3390/vaccines10040575

**Published:** 2022-04-08

**Authors:** Bethlehem Mengesha, Asen G. Asenov, Bruria Hirsh-Raccah, Offer Amir, Orit Pappo, Rabea Asleh

**Affiliations:** 1Heart Institute, Hadassah Medical Center, Faculty of Medicine, Hebrew University of Jerusalem, Jerusalem 9112001, Israel; bettytade@gmail.com (B.M.); asen.g.asenov@gmail.com (A.G.A.); bruria.hirsh@mail.huji.ac.il (B.H.-R.); oamir@hadassah.org.il (O.A.); 2Division of Clinical Pharmacology, Institute for Drug Research, School of Pharmacy, Faculty of Medicine, Hebrew University of Jerusalem, Jerusalem 9112001, Israel; 3Department of Pathology, Hadassah Medical Center, Faculty of Medicine, Hebrew University of Jerusalem, Jerusalem 9112001, Israel; lorit@hadassah.org.il

**Keywords:** myocarditis, COVID-19, mRNA vaccination, booster

## Abstract

Vaccination with mRNA vaccines against coronavirus disease 2019 (COVID-19) has been associated with a risk of developing myocarditis and pericarditis, with an estimated standardized incidence ratio of myocarditis being 5.34 (95% CI, 4.48 to 6.40) as compared to the expected incidence based on historical data according to a large national study in Israel. Most cases of myocarditis in vaccine recipients occur in young males, particularly following the second dose, and the presentation is usually mild. Recently, the third (booster) dose has been shown to reduce confirmed infections and severe illness even against common variants of the virus. In Israel, over 4.4 million citizens (more than 45% of the population) have been vaccinated with the third dose of Pfizer-BioNTech vaccine BNT162b2. Herein, we report the first case of a histologically confirmed severe myocarditis following the third dose of BNT162b2 COVID-19 vaccine.

## 1. Patient Presentation

A 43-year-old female with no past medical history presented with palpitations and shortness of breath started 2 days after receiving the third dose of BNT162b2 COVID-19 vaccine. No side effects of the vaccine were noted following the preceding two doses received five months prior. On presentation, her electrocardiogram showed rapid monomorphic ventricular tachycardia (VT) requiring a synchronized cardioversion (Figure 1). Physical examination findings were unremarkable.

Laboratory blood tests showed normal complete blood count, elevated C-reactive protein up to 6.7 mg/dl, and high-sensitive troponin I up to 2082 ng/L. D-dimer was also elevated at 1.3 ng/L, but a Chest CT angiography ruled out acute pulmonary embolism. Polymerase-chain-reaction (PCR) tests of nasopharyngeal swab obtained on admission for severe acute respiratory syndrome coronavirus 2 (SARS-CoV-2) and for other viruses causing upper respiratory tract infection was negative. A comprehensive autoimmune profile testing in blood, including autoantibodies for the sarcomeric protein titin were all unremarkable (Table 1A).

A transthoracic echocardiogram showed mild-to-moderate left ventricular (LV) systolic dysfunction with ejection fraction of 40–45% with anterolateral and inferolateral wall motion abnormalities and normal right ventricle systolic function. Cardiac magnetic resonance imaging (MRI) showed myocardial edema, demonstrated by increased regional T2 signaling, and transmural late gadolinium enhancement (LGE) of the inferolateral wall on LGE imaging (Figure 2). 

Coronary angiography showed normal coronary arteries (Figure 1). Right heart catheterization showed borderline normal filling pressures and normal cardiac index of 3.1 L/min/m^2^. An endomyocardial biopsy specimen taken form the right ventricle showed mononuclear infiltration with destruction of muscle fibers confirming the diagnosis of acute myocarditis (Figure 3). The endomyocardial specimen was negative for enterovirus, herpes viruses, parvovirus, adenovirus, SARS-Cov-2, and other respiratory viruses. Furthermore, a genetic counselling and whole exome sequencing found no pathogenic variants causing cardiomyopathy. 

## 2. Hospital Course

The patient was treated with prednisone (1 mg/kg, with gradual dose tapering over the next 3 months), bisoprolol, and ramipril, with complete resolution of symptoms, inflammatory markers, and troponin elevation. Her LV systolic function remained mildly to moderately reduced, and given her presentation with life-threatening arrhythmia, the patient was discharged home with a wearable defibrillator (LifeVest) while on steroid therapy. 

## 3. Follow-Up Course

Two weeks after discharge, the patient was readmitted after receiving an appropriate shock from the LifeVest with a device interrogation demonstrating a sustained monomorphic VT event (Figure 4 and Appendix A). Troponin and inflammatory markers in blood were normal, and echocardiographic findings were unchanged. The beta-blocker dose was increased, and the patient was discharged home with a recommendation to continue steroid therapy while on a wearable defibrillator. 

Three months after her initial presentation, the patient remained asymptomatic after steroid discontinuation, and her subsequent course was uneventful. Cardiac MRI showed a resolution of myocardial edema but a persistent transmural LGE in the inferolateral wall, consistent with residual myocardial scar, with a mildly to-moderately reduced LV systolic function (Appendix A). Moreover, a repeated endomyocardial biopsy showed mild interstitial fibrosis with resolution of inflammation without evidence of cardiomyocyte damage (Appendix A). 

## 4. Discussion

Vaccine-associated myocarditis (VAM) is a rare entity. Although myocarditis and pericarditis have been reported following non-COVID-19 vaccines, such as smallpox, influenza, anthrax, typhoid, and HBV vaccines, the incidences were much higher following COVID-19 vaccines [8]. Although this is most likely due to the large population receiving the COVID-19 vaccine, other possible mechanisms need to be examined.

Although several hypotheses have been proposed, the exact mechanism in which myocardial inflammation occurs is unknown. Some of the proposed mechanisms include molecular mimicry between the spike protein of SARS-CoV-2 and cardiac self-antigen, resulting in aberrant innate and acquired immune system responses.^2^ Furthermore, it has been suggested that mRNA vaccines may emulate the effects of live vaccines, and that interferon-gamma production by type 1 T helper (Th1) cells following mRNA vaccination may play a role in cardiac inflammation [8]. 

Most cases of myocarditis following COVID-19 vaccine occur in young males, particularly following the second dose, and the presentation is usually mild [1,2,3,4,5]. The higher incidence among males is unclear but may be due to different effects of sex hormones on the cell-mediated immune response, with testosterone promoting a more pronounced Th1 stimulation, whereas estrogen displays inhibitory effects on pro-inflammatory T cells [9]. 

Recently, the third (booster) dose has been shown to reduce confirmed infections and severe illness even against common variants of the virus [6]. In Israel, approximately half of the population (over 4.4 million citizens) received the third vaccination dose of Pfizer-BioNTech vaccine BNT162b2 [7]. In this article, we report the first case of a histologically confirmed severe acute myocarditis following the third dose of BNT162b2 COVID-19 vaccine. Myocarditis following the third (booster) dose to our knowledge has not been reported to date, and the rarity of this condition may be attributed to the fact that patients who develop myocarditis after the first or second dose of mRNA vaccines, perhaps due to immunologic or/and genetic susceptibility, are prohibited from receiving the booster dose of these mRNA vaccines. Therefore, estimating the risk of myocarditis following the third (booster) dose could be challenging.

Several differential diagnoses have been considered in our patient; hence, a thorough investigation has been performed to rule out coronary artery disease, viral infections, and genetic and autoimmune diseases. Although a definitive causal relationship cannot be determined, the proximity to the COVID-19 mRNA vaccine administration accompanied by the lack of other etiologies make VAM the most likely diagnosis. 

Obesity is associated with chronic inflammation and has been linked with severe illness and increased in-hospital mortality among patients with COVID-19 infection [10]. As a result, vaccination is highly advised in this at-risk population. Although data are scarce concerning COVID-19 vaccine response among obese individuals, some studies show lower antibody titers among obese individuals compared to those with a normal or low body mass index (BMI), whereas other studies show no difference in vaccine efficacy among individuals with different BMIs [11,12,13]. Our patient has no chronic medical condition except for obesity, which raises the question whether obesity plays a role in the pathophysiology of VAM. Iguacel et al. [13] investigated the relationship between weight status and severity of side effects from COVID-19 vaccine and found significantly higher adverse events (fever, vomiting, diarrhea, and chills) in those who were not overweight compared to those overweight. Consequently, there are no clear data suggesting association between obesity and an increased risk of VAM in the literature [11,12,13].

The diagnosis of myocarditis was definitive based on the histological findings. However, the cardiac MRI findings were atypical in that a transmural LGE with a thin and aneurysmatic lateral and inferolateral wall of the LV were observed. Although rare, atypical presentations of myocarditis with transmural or diffuse LGE have been described in the literature [14]. One possible explanation in these scenarios may be coronary spasm, which seldom accompanies acute myocarditis, resulting in myocardial ischemic damage. This entity was not witnessed in our patient during coronary angiography. Conversely, this atypical MRI findings could be a marker of myocardial inflammation severity. As part of the differential diagnosis, inflammatory cardiomyopathies such as cardiac sarcoidosis and giant cell myocarditis were considered. Neither inflammatory noncaseating granulomas nor multinucleated giant cells were seen in histopathology on repeated EMB. Moreover, a ^18^F-flourodeoxyglucose (FDG) positron emission tomography (PET) scan was performed for this patient, and it did not show any evidence of systemic sarcoidosis or isolated cardiac sarcoidosis. 

Few case reports on VAM following COVID-19 vaccination describe histopathologic findings from EMB or autopsy results [15,16,17,18]. Some of these cases describe areas of acute inflammation with myocyte necrosis with predominantly lymphocytic infiltration similar to our patient, whereas others describe diffuse or scattered foci of contraction band necrosis with hypercontracted sarcomeres and a predominantly neutrophil and histocyte infiltrates. The pattern of injury in some of these cases may suggest catecholamine-induced myocardial inflammation as the mechanism underlying this condition [16]. In another study where EMB and autopsy were performed, an inflammatory infiltrate, predominantly composed of T-cells and macrophages, admixed with eosinophils, B cells, and plasma cells, were seen [18]. The heterogeneity if these histopathologic findings may suggest various mechanisms of myocardial injury are at play in the pathophysiology of VAM.

Since most cases of myocarditis following the mRNA vaccine have mild clinical course and are self-limiting, a specific treatment regimen has not been proposed. In a retrospective observational study of 136 adolescents and young adults with clinically suspected VAM, the majority (91%) of patients received anti-inflammatory treatments, including non-steroidal anti-inflammatory drugs (NSAIDS), intravenous immunoglobulin (IVIG), glucocorticoids, or colchicine, to curb the exaggerated immune-mediated responses [19]. However, due to the small sample size, no definitive conclusions can be drawn regarding the efficacy of specific anti-inflammatory agents in the setting of VAM. Due to the scarcity of data, the appropriate choice of medical treatment may be challenging, although in the rarer cases of severe myocarditis, more aggressive immunosuppressive therapies (such as high dose steroids as in the present case) are apparently needed to block the immune-mediated myocardial injury related to the mRNA vaccine. 

## 5. Conclusions

In the current case, we first describe histologically confirmed severe myocarditis developed 2 days after the third dose of BNT162b2 COVID-19 vaccine. Apart from obesity, the patient was healthy with no concomitant diseases and no adverse reactions after the first and second vaccination. Although a direct causal link with the vaccine cannot be confirmed, extensive viral, autoimmune, and genetic investigation, both in the blood and myocardial tissues, ruled out other etiologies for myocarditis. VAM following COVID-19 vaccination is a rare condition and generally presents as a mild and self-limiting illness. On the contrary, COVID-19 infection is implicated in various extracardiac and cardiac complications, including severe myocardial injury, which is more common and severe than those observed with the mRNA vaccination [20]. Therefore, the benefit–risk assessment is obviously in favor of receiving the vaccine for COVID-19 due to the exceedingly greater advantages for preventing severe illness, including COVID-19-induced myocarditis. Although several hypotheses have been proposed, a clear understanding of the mechanism underlying VAM condition is still lacking. More studies are needed to provide further insights into the underlying mechanisms and possible prevention of this rare adverse effect.

## Figures and Tables

**Figure 1 vaccines-10-00575-f001:**
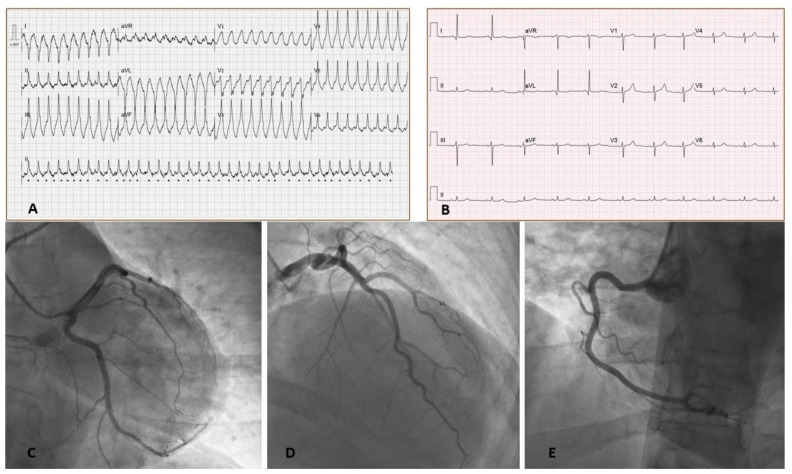
Electrocardiograms (EKG) on presentation. (**A**) EKG on presentation showing monomorphic ventricular tachycardia. (**B**) EKG after cardioversion showing sinus rhythm with poor R wave progression in precordial leads without significant PR or ST segment changes. Left heart Catheterization (**C**–**E**) show normal coronary arteries. (Table 1B).

**Figure 2 vaccines-10-00575-f002:**
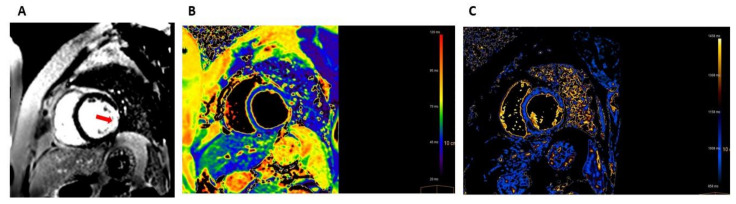
Cardiac MRI (1.5T) at presentation. (**A**) A short axis view of late gadolinium enhancement (LGE) image showing transmural LGE in the lateral and inferolateral wall (red arrow). (**B**) A T2 mapping image showing myocardial edema with regionally increased T2 values (78 milliseconds) in the lateral and inferolateral wall. (**C**) A T1 mapping image showing myocardial injury with increased native T1 values (1288 milliseconds) in the lateral and inferolateral walls.

**Figure 3 vaccines-10-00575-f003:**
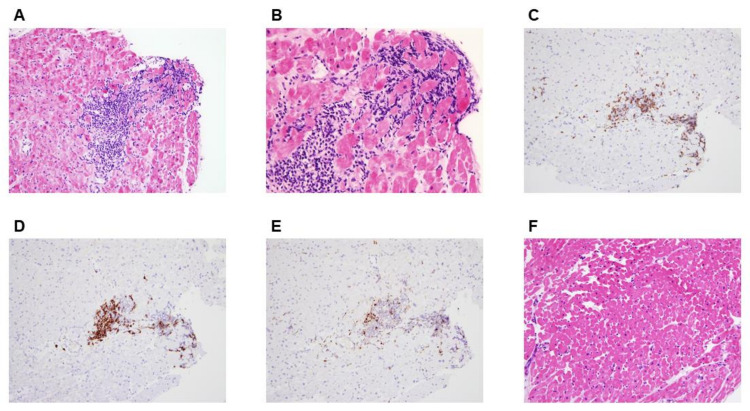
Histopathological Findings from Endomyocardial Biopsies. Hematoxylin-eosin stains of heart-tissue specimens showed inflammatory infiltrates composed of mononuclear cells with foci of severe interstitial edema and multifocal cardiomyocyte damage consistent with the diagnosis of definitive acute myocarditis (Panels **A** and **B**). Immunostaining demonstrated predominant inflammatory infiltration of lymphocytes (Panel **C**, CD3 immunostaining for T-lymphocytes, and Panel **D**, CD20 immunostaining for B-lymphocytes) and, to a lesser extent, macrophages (Panel **E**, CD68 immunostaining). A repeated endomyocardial biopsy 3 months later showed a complete resolution of myocarditis without evidence of inflammation or cardiomyocyte damage (Panel **F**, hematoxylin-eosin stains). Original magnification, ×200; scale bars, 100 µm (Panels **A**, and **D**–**F**) and original magnification, ×400; scale bars, 50 µm (Panel B).

**Figure 4 vaccines-10-00575-f004:**
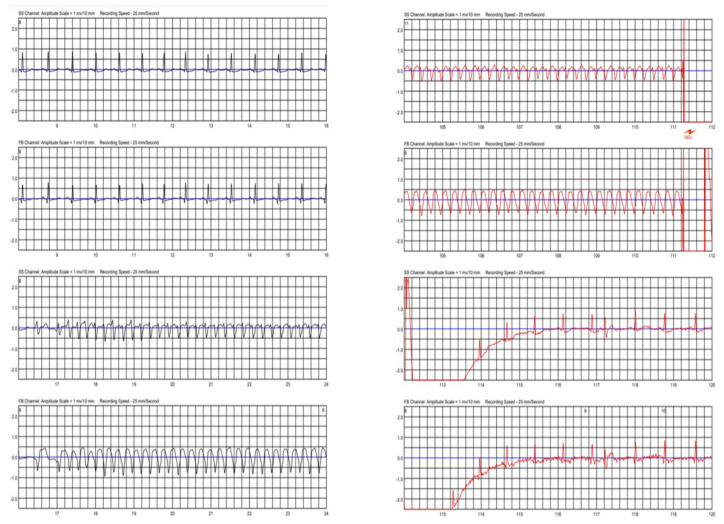
Two-lead recording from a wearable cardioverter-defibrillation (LifeVest) demonstrating a sustained monomorphic ventricular tachycardia event lasting for 94 s, which was ultimately terminated by a successful electrical shock, about 2 weeks after the index hospitalization. The figure illustrates the onset (time 17 s) and termination of the ventricular tachycardia event (time 111 s).

**Table 1 vaccines-10-00575-t001:** Patient characteristics, clinical investigations and medical therapy.

A: Clinical Characteristics and Laboratory Findings General Characteristics	
Age	43
Sex	Female
Race	White, Arab Israeli
Medical history	Obesity
Medications prior to illness	None
Admission vital signs	
Heart rate (bpm)	116
Respiratory rate (per minute)	20
Blood pressure (mmHg)	102/86
Temperature (°C)	37.3
Pulse oximeter saturation (%)	96
Body Mass Index (kg/m^2^)	33.5
Laboratory findings	
Admission Laboratories	
WBC (×10^3^/µL)	8.8
Neutrophils (%)	69.1
Lymphocytes (%)	21.9
Eosinophils (%)	2.5
Hemoglobin (g/dL)	12.8
Hematocrit (%)	40.3
Platelets (×10^3^/µL)	315
Sodium (mmol/L)	136
Potassium (mmol/L)	3.6
BUN (mg/dL)	21
Creatinine (mg/dL)	0.8
AST (U/L)	42 (reference: 8–48 ng/L)
ALT (U/L)	51 (reference: 7–55 ng/L)
Alkaline phosphatase (U/L)	57 (reference: 40–129 ng/L)
Total bilirubin (mg/dL)	7.5 (reference: 0.1–1.2 mg/dL)
Troponin I (high-sensitive) (ng/L)	630 (reference: 0–34 ng/L)
Creatinine phosphokinase (U/L)	97 (reference: 26–192 U/L)
Other labs	
Peak troponin I (high-sensitive) (ng/L)	2082 (reference: 0–34 ng/L)
Lactate	2.2 (reference: 0.5–2.2 mmol/L)
TSH (mIU/L)	2.1 (reference: 0.3–4.2 mIU/L)
C-reactive protein (mg/dL)	6.8 (reference: <0.5 mg/dL)
Ferritin (ng/mL)	105 (reference: 10–291 ng/mL)
NT-proBNP (pg/mL)	257 (reference: 0–125 pg/mL)
PT (%)	100 (reference: 63.8–127.7)
aPTT (sec)	28.7 (reference: 22.4–36.3)
INR	1.0 (reference: 0.9–1.2)
D-dimer (mg/L)	1.3 (reference: 0–0.4 mg/L)
Autoimmune evaluation	
p-ANCA	Negative
c-ANCA	Negative
ANA	Negative
IgA (mg/dL)	190.8 (reference: 70–400 mg/dL)
IgM (mg/dL)	160.8 (reference: 40–230 mg/dL)
IgG (mg/dL)	596 (reference: 700–1700 mg/dL)
C3 (mg/dL)	207.6 (reference: 90–180 mg/dL)
C4 (mg/dL)	41.8 (reference: 10–40 mg/dL)
Anti-Titin autoantibodies	Negative
Viral evaluation in nasopharyngeal swab specimens (PCR)	
SARS-CoV-2 PCR	Negative
Respiratory viral panel (PCR):	
Adenovirus	Negative
Human Metapneumovirus	Negative
Rhinovirus/Enterovirus	Negative
Influenza virus type A (non H1N1)	Negative
Influenza virus type A (H1N1)	Negative
Influenza virus type B	Negative
Parainfluenza virus types 1/2/3	Negative
Respiratory Syncytial Virus (RSV)	Negative
Viral serology (antibodies in serum)	
COVID-19 (IgM) *	Negative
COVID-19 (Anti S IgG) *	Positive (14,135 AU/mL), (reference: <50 AU/mL)
COVID-19 (Anti N IgG) *	Negative
Adenovirus	Negative
CMV	Negative
EBV	Negative
Enterovirus	Negative
HHV-6	Negative
HSV-1	Negative
HSV-2	Negative
Myocardial tissue specimens (PCR)	
Enterovirus	Negative
Adenovirus	Negative
Human Metapneumovirus	Negative
Parvovirus	Negative
HHV-6	Negative
Influenza virus type A (non H1N1)	Negative
Influenza virus type A (H1N1)	Negative
Influenza virus type B	Negative
Parainfluenza virus types 1/2/3	Negative
Respiratory Syncytial Virus (RSV)	Negative
SARS-Cov-2	Negative
**B: Cardiac Diagnostic Testing and Medical Treatment**	
ECG	Initial presentation: Rapid monomorphic ventricular tachycardia with ventricular rate of 120 bpm.
	After cardioversion: Normal sinus rhythm, ventricular rate of 64 bpm, normal axis, no ST-T or PR changes, no QT prolongation, and normal QRS duration.
Transthoracic echocardiogram	Mildly to moderately decreased LV systolic function, EF 40–45%, mildly increased LV size (mid LV diameter of 5.8 cm), normal RV size and function, normal RA and LA size, mild MR and TR, normal IVC size (<2.1 cm), no pericardial effusion
Cardiac MRI	Mildly to moderately reduced systolic function, hypokinesia of the lateral wall and akinesia of an aneurysmatic inferolateral wall, myocardial edema in the inferolateral wall, subendocardial and transmural LGE in the lateral wall and inferolateral wall.
Coronary angiography	Right dominant system
	No angiographic evidence of CAD
Hemodynamics/right heart catheterization	RAP 4 mmHg
	RVP 30/5 mmHg
	PAP 30/15 mmHg, mPAP 20 mmHg
	PCWP 14 mmHg
	AOP 122/70 mmHg
	TPG 6
	PVR 1.0
	CO 6.0 L/min (per Fick)
	CI 3.1 L/min/m^2^ (per Fick)
Medications received during illness/hospitalization	Prednisone
	Omeprazole
	Bisoprolol
	Ramipril
	Sulfamethoxazole/Trimethoprim

* Given negative PCR testing in the absence of IgM and anti-nucleocapsid (anti-N) antibodies for COVID-19, detectable anti-surface (anti-S) antibodies for the virus indicates vaccination for COVID-19 and not an active infection or prior exposure to the virus. Abbreviations: bpm, beats per minute; RV, right ventricle, LV, left ventricle; EF, ejection fraction; MR, mitral regurgitation; TR, tricuspid regurgitation; RA, right atrium; LA, left atrium; IVC, inferior vena cava; LGE, late gadolinium enhancement; CAD, coronary artery disease; RAP, right atrial pressure; RVP, right ventricular pressure; PAP, pulmonary artery pressure; mPAP, mean pulmonary artery pressure; PCWP, pulmonary capillary wedge pressure; AOP, aortic pressure, TPG, transpulmonary gradient; PVR, pulmonary vascular resistance; CO, cardiac output; CI, cardiac index.

## Data Availability

All data needed to evaluate the conclusions in the article are present in the manuscript.

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
