# Peer review of "Severe Acute Myocarditis after the Third (Booster) Dose of mRNA COVID-19 Vaccination"

_vaccines, 2022, doi:10.3390/vaccines10040575_

Round 1

Reviewer 1 Report

Some comments to correct:

Lines 14 and 15: In this point, should be demonstrated with "strong data"  the risk of myocarditis or pericarditis (OR value o HR value observed in other studies).

Lines 31-33: It would be interesting know the coagulation parameters of the patient. A pro-coagulative state could be a remarkable point in this case.

Table 1: This table is too long. Is necessary a reorganization the data. My suggestion is a divide between laboratory findings and Clinical Findings. In the section laboratory, including coagulation parameters. If the respiratory virus panel is negative, I recommend present in a supplementary table.

Lines 103-104: The patient presents a BMI of 33. This point not be forgotten and should be remarkable in this article. Exists many articles that describe the effect of obesity on the side effects of vaccines.

 https://doi.org/10.1002/dmrr.3465

https://doi.org/10.1007/s11695-021-05404-y

Discussion: In the same way as before comment, is important to include the participation of obesity in the side effects of Covid-19 vaccine. The biological interaction of obesity and the vaccine, more the molecular mimicry of cardiac self-antigen, is an important point to discuss.

Lines 155-160: I recommend re-written the conclusion paragraph for more clarity. It´s important to highlight the strong points of this case ( a complete study of many causes of myocarditis and pericarditis, for eliminated possibles confounding factors), and the limitations of this case.

Author Response

We highly appreciate the important comments raised by the Reviewers. We have provided point-by-point responses to their comments and modified the Article accordingly.  

Reviewer 1

Lines 14 and 15: In this point, should be demonstrated with "strong data" the risk of myocarditis or pericarditis (OR value o HR value observed in other studies).

RE: We appreciate the Reviewer`s comment. In the revised manuscript, we have added the following data to the introduction: “Vaccination with mRNA vaccines against coronavirus disease 2019 (Covid-19) has been associated with a risk of developing myocarditis and pericarditis, with an estimated standardized incidence ratio of myocarditis being 5.34 (95% CI, 4.48 to 6.40) as compared to the expected incidence based on historical data according to a large national study in Israel.

Lines 31-33: It would be interesting to know the coagulation parameters of the patient. A pro-coagulative state could be a remarkable point in this case.

RE: We thank the Reviewer for this excellent point. We have included the coagulation profile in Table 1 as requested. As mentioned in the main text, the D-dimer was interestingly elevated in this patient at presentation, but CT angiography of chest was performed, and it ruled out pulmonary embolism. The other coagulation parameters were within normal range.

Table 1: This table is too long. Is necessary a reorganization the data. My suggestion is a divide between laboratory findings and Clinical Findings. In the section laboratory, including coagulation parameters. If the respiratory virus panel is negative, I recommend present in a supplementary table.

RE: We appreciate the Reviewer`s comment. As suggested, we have rearranged Table 1 and divided it into 3 different sections to include demographic, laboratory findings and clinical findings. Although a comprehensive evaluation for viral etiology, including in the myocardial tissues, was negative, we believe that this information is a pertinent finding and mandatory for the exclusion of viral myocarditis. Therefore, we respectfully ask to keep these data in Table 1 within the main text. 

Lines 103-104: The patient presents a BMI of 33. This point not be forgotten and should be remarkable in this article. Exists many articles that describe the effect of obesity on the side effects of vaccines.

 https://doi.org/10.1002/dmrr.3465

https://doi.org/10.1007/s11695-021-05404-y

RE: We thank the Reviewer for this excellent comment. Previous studies have shown that obesity is a significant predictor of severe COVID-19 and mortality from the disease. However, the data in literature regarding the implication of obesity in vaccine-associated myocarditis (VAM) or other post-vaccination adverse events are scarce and based on small observational studies. Interestingly, some studies have suggested lower antibody titers and lower adverse events (such as fever, vomiting, and chills) post vaccination among obese individuals. Further studies are warranted to investigate whether obesity has significant effect on the development of myocarditis or other adverse effects. We have elaborated further on this topic and included relevant references in the discussion section as suggested.

Discussion: In the same way as before comment, is important to include the participation of obesity in the side effects of Covid-19 vaccine. The biological interaction of obesity and the vaccine, more the molecular mimicry of cardiac self-antigen, is an important point to discuss.

RE: This is an excellent point. We have elaborated on this further in the discussion section and included the currently available data in literature regarding the association of obesity with vaccine-related side effects as well as potential underlying mechanisms of this association.

Lines 155-160: I recommend re-written the conclusion paragraph for more clarity. It´s important to highlight the strong points of this case (a complete study of many causes of myocarditis and pericarditis, for eliminated possible confounding factors), and the limitations of this case.

RE: We appreciate the Reviewer`s comment. We have revised the conclusion section and summarized the most important points of this case and the main massage to the readership.

Reviewer 2 Report

This is a well written and timely case of acute myocarditis following the third SRAS-CoV-2 mRNA vaccination. The methods are described in a comprehensive fashion and provide valubale isnights to this rare but important condition. The findings are critically discussed, and I do not have any comments or suggestions.

Author Response

This is a well written and timely case of acute myocarditis following the third SRAS-CoV-2 mRNA vaccination. The methods are described in a comprehensive fashion and provide valuable insights to this rare but important condition. The findings are critically discussed, and I do not have any comments or suggestions.

RE: We thank the reviewer for these positive comments.

Reviewer 3 Report

Gill JR and colleagues have recently described histopathologic characteristics of hearts from 2 subject died after the vaccine. Despite this was the second dose, there are certain parallelism, which should be discussed in the article (Gill JR, Tashjian R, Duncanson E. Autopsy Histopathologic Cardiac Findings in Two Adolescents Following the Second COVID-19 Vaccine Dose. Arch Pathol Lab Med. 2022 Feb 14. doi: 10.5858/arpa.2021-0435-SA).

Author Response

Gill JR and colleagues have recently described histopathologic characteristics of hearts from 2 subject died after the vaccine. Despite this was the second dose, there are certain parallelism, which should be discussed in the article (Gill JR, Tashjian R, Duncanson E. Autopsy Histopathologic Cardiac Findings in Two Adolescents Following the Second COVID-19 Vaccine Dose. Arch Pathol Lab Med. 2022 Feb 14. doi: 10.5858/arpa.2021-0435-SA).

RE: We thank the Reviewer for this valuable comment. As suggested, we have added additional histopathologic data on vaccine-associated myocarditis (VAM), including the suggested reference and others.